

# On the Use of Measurements from a Commercial Microwave Link for Evaluation of Flash Floods in Arid Regions

Adam Eshel[1], Hagit Messer[2], Jonatan Ostrometzky[2], Roi Raich[2], Pinhas Alpert[1], and Jonathan B. Laronne[3]

[1]Department of Geophysics, Tel Aviv University, Tel Aviv, Israel
[2]School of Electrical Engineering, Tel Aviv University, Tel Aviv, Israel
[3]Department of Geography and Environmental Development, Ben Gurion University, Beer Sheva, Israel

*Correspondence to:* Adam Eshel (adameshel@mail.tau.ac.il)

**Abstract.**

Flash flood warning in deserts is a challenging task, and local rain bursts are of high significance. In the last decade, commercial microwave telecommunication links have been shown to be a valuable tool for near ground rainfall estimations. Dense networks are used for highly accurate rainfall estimates, while isolated links, typical in rural areas, can detect the existence of

near-ground rainfall. However, the induced attenuation of the signal integrates the rainfall along a line, therefore high local rain intensities are smoothed. In this paper, we propose a novel method that uses the integration of measurements from an isolated long microwave link with measurements from weather radar to identify potential conditions for flash floods. In particular, we suggest using radar measurements to indicate the rain variability (spottiness) along a 16 km long link, crossing Wadi Ze'elim catchment ($245 \text{ km}^2$), using kurtosis as a spottiness index. Quantitative ground level rainfall measurements are then provided

by the link. When compared with analyzed discharge, inverse kurtosis-rain rate relation is associated with flash flood responses in Wadi Ze'elim, an ephemeral Wadi located in the Dead Sea.

## 1 Introduction

Flash floods are a common phenomenon in arid regions. They serve as the main source of water to support the existence of flora and fauna, and play an important role in ground water recharge through transmission losses. The flood events can be

generated by both very short and longer duration rainfall events, yet high local and temporal rain intensities, which are much less predictable (David et al., 2009), are more likely to be the cause for the abrupt floods. When civilization meets areas prone to flash floods, severe damage to infrastructure, roads, and private property can frequently result, and therefore, a threat to human life is inevitable. This form of runoff is typical of desert areas, because of low infiltration rates mainly due to poor vegetation coverage, shallow underdeveloped rocky soils, exposed bedrock, and steep rocky terrain. The dependence of surface runoff on

rainfall characteristics in such areas, particularly spatial distribution and rain intensity, is rather strong (Pilgrim et al., 1988). Effective rainfall monitoring is therefore crucial for understanding the processes that take place when these floods occur.



## 1.1 Rainfall Monitoring

Since rain gauges provide measurements from ground level, their values are justifiably referred to as the "ground truth." Arid regions, as well as mountainous ones, are known to have particularly low density rain gauge networks (Pilgrim et al., 1988; Marra et al., 2014). Furthermore, the resulting data are point measurements, and as flash flood generating rainfall events are
5 commonly highly localized (spotty), the dearth of areal coverage generates large uncertainties in the identification of rainfall patterns and local high intensities (David et al., 2013).

Weather radar is an additional widely used tool for rain monitoring. Offering a wide spatial coverage for various elevations and a temporal resolution of minutes, it provides indirect rain measurements of crucial value. Rainfall measurement accuracy is of high importance in surface hydrology modeling (Larson and Peck, 1974; Osborn et al., 1979; Samuels et al., 2011;
10 Price et al., 2014). It has been shown that the prediction of flash floods is improved when complementary weather radar, rather than rain gauges alone, is used (Berne and Krajewski, 2013; Morin et al., 2009). DDespite this, weather radars are used less extensively for hydrological forecasting than expected because of the need for adjustment and various sources of errors (Joss et al., 1990), arising mostly as a result of the large distances of the radar station from the monitored locations, robust working premises (Harrison et al., 2000), ray blocking obstacles (Alpert and Shafir, 1989), observations at high altitudes, and
15 the evaporation of water droplets (Price et al., 2014). In particular, mountainous areas induce additional factors that affect the accuracy of radars (Germann et al., 2006; Alpert and Shafir, 1989).

Monitoring rain through the use of Commercial (telecommunication) Microwave Links (CMLs) attenuation data of existing wireless communication systems, first presented more than a decade ago (Messer et al., 2006; Leijnse et al., 2007), is increasingly becoming a legitimate tool in meteorological sciences (Alpert et al., 2016; Overeem et al., 2016), and it has been shown
20 that it is feasible to obtain data in real time Chwala et al. (2015). A radio signal, which operates in either the K-band or the E-band frequencies, is sensitive to a number of environmental phenomena, of which rain is the dominant one (ITU-R, 2009). The transmitted signal from one tower is received by a nearby one, i.e., the receiver, and in the absence of technical or climatic disturbances, the differences between the transmitted signal level (TSL) and the received signal level (RSL), given in dB, are almost constant and depend mainly on the fixed distance between the towers. When rain falls within the medium the signal
25 power level weakens resulting in lower RSLs than in dry periods, from which the close to ground-level line integrated rain intensities can be derived. This relation between the signal attenuation level and the specific rain intensity has been studied (Gunn and East, 1954) and found to follow a logarithmic relationship in the form of a *Power Law* (Olsen et al., 1978).

The feasibility of reconstructing 2D rain maps using the recorded TSL and RSL values was previously shown (Messer et al., 2008; Overeem et al., 2013; Liberman and Messer, 2014). The benefit of interpolated spatial CML Rainfall (CMLR), as well
30 as data obtained from a single link, has previously been shown (Rayitsfeld et al., 2012). However, the performance of the aforementioned methodologies depends largely on the density of the network. Therefore, CMLs can provide a fair ground truth for rain in populated areas, where the networks are denser. In contrast, remote rural areas are frequently characterized by low density commercial microwave networks, i.e., fewer and longer links. The detection of rain cells by both rain gauges and CMLs in the arid and remote southern part of Israel was previously studied (David et al., 2013). The paper presented the benefits of



the presence of long links in sparsely populated areas, stating that the links detect rain cells in more cases and earlier than gauges.

## 1.2 Background

Spotty rainfall events are common worldwide, especially in arid regions, such as the Judean Desert and Dead Sea, and the difficulty in their monitoring is of considerable significance because of the related high potential rain intensities (Amiran, 1995; Shentsis et al., 2012). An earlier study, which addressed watershed hydrological responses to spatial patterns of rainfall in a semi-arid area, suggested that, at a distance of $2.1-3.2$ km from the rain cell center (where the cell's maximal rain intensity is present), the rain intensity is reduced dramatically Morin et al. (2006). Furthermore, defined as a rain shadow region, the Judean Desert is characterized by very localized rain cells, possibly also because of the low availability of moisture in the desert air Sharon (1972). Rainfall events, which occur several times a year, frequently generate flash floods in the region (Kahana et al., 2002; Tarolli et al., 2012; Ziv et al., 2006).

The integration of CML data in the world of hydrology is not trivial for various reasons, one of which stands out: the resolution in space of quantitative precipitation estimates is of high importance for hydrologic forecasting, as heavy rain generates runoff faster than light rain. Rain estimates derived from CML data are, in practice, a line integrated rain intensity between two towers. Since the radio signal is affected by rain properties within its medium, there is a slight to non-existent difference between the attenuation induced by heavy but spotty rainfall and that of uniformly spread but light rainfall. Each measurement can be referred to as the average rain intensity along the path, and as a result, similar attenuation patterns can result in the two different cases, thus leading to similar CMLR. Considering the typical size of rain cells in semi-arid areas and the typical large length of CMLs in non-urban ones (several-tens of kilometers), the provision of suitable data for surface hydrology, derived solely from CMLs, is challenging.

The goal of this study was to propose a novel integration of measurements from two existing instruments, weather radar and CMLs, to improve understanding of flash flood generation and potentially to allow flash flood warning. The former instrument is employed to indicate spatial rain variability, i.e., rainfall spottiness, while the near-ground measurements provided by the latter contribute to the accuracy of the rainfall estimates. The approach is a complementary integration, using the advantages of each rain monitoring instrument to compensate the weaknesses of the other, with respect to the hydrological responses measured at the outlet. Results for Wadi Ze'elim in the Dead Sea area are shown. Five storms with different spottiness properties were analyzed. Four rain gauges were used for testing our proposed algorithm and for validation.

## 2 Study Area

The study region sprawls along the western part of the Dead Sea rift and is focused on latitudes around $31.3^o$. The characteristic surface comprises mostly limestone, dolomite chalk, and cherts. The arid to hyper-arid climate, extreme summer temperatures, and the steep topography are some of the causes of the sparse distribution of the population. Hence, the spread of the cellular infrastructure is mostly along roads.



Wadi Ze'elim is a 245 km$^2$ catchment (up to the hydrometric station located at the fan apex, Fig. 1) draining the eastern slopes of the Judean Desert to the Dead Sea ($-430$ m below sea level). East of the divider (the Judean mountains), an area with very moderate slopes, is present, the desert plateau. Desert lithosols and regs characterize this upper (western) part of the basin (Cohen and Laronne, 2005). A stony surface and exposed limestone bedrock commonly occur in the steep slopes of the lower

(eastern) part, together with a similar soil texture on the floodplains (Yair and Lavee, 1976). Topography is generally rugged and waterfalls are prevalent. Vegetation coverage is rare, discontinuous, and occurs mostly in channels.

The annual rainfall gradient is very sharp in the region because of the rain-shadow effect: the northwestern part of the catchment tangents 300 mm y$^{-1}$, whereas the Dead Sea shore, considered a hyper-arid zone, averages less than 50 mm y$^{-1}$ (Morin et al., 2009). The horizontal distance between the uppermost part and the outlet of the catchment is only about 20 km

(Fig. 1). This implies that most of the annual rainfall is recorded by gauges located in the western part of the basin, as is the studied CML.

The synoptic systems dominating the rain events in the region are the Red Sea trough and the winter cyclones (Mediterranean Lows) (Krichak et al., 2000; Kahana et al., 2002). The first is a synoptic scale system, active in the region mostly during spring and characterized by convective events, typified by low correlations between nearby rain gauges (Osborn et al., 1979), which

leads to 31% of the flash floods in southern Israel (Shentsis et al., 2012). The latter frequently coincide with cold fronts, arriving from the Mediterranean Sea in the west, which, if conditions allow, generate rain on the lee side of the Judean Mountains, intermittently causing the eastern drainage system of the Dead Sea to flow.

## 3  Data Analysis

### 3.1  Hydrology

The trunk channel of Wadi Ze'elim was monitored continuously at a hydrometric station located in close proximity to road 90, parallel to the Dead Sea shore. The longitudinal channel profile of Wadi Ze'elim, from the CML channel crossing point to the hydrometric station at the outlet, is shown in Fig. 3 and was used later to estimate the travel time of a flood wave.

Mean flow velocity was calculated based on rare measurements of average surface water velocity using a Doppler-based hand-held radar velocimeter (Welber et al., 2016). The discharge calculations were in the form

$$Q = v \cdot A^f \quad (\text{m}^3\ \text{s}^{-1}) \tag{1}$$

where $v$ is the average surface water velocity (in m s$^{-1}$) and $A^f$ is the cross-sectional area of flow (in m$^2$). Velocity measurements were possible on two occasions. Water depth was monitored by a pressure transducer, located at the height of the river bed, and therefore, the determination of water depth also during shallow recessions was possible.



## 3.2 Microwave Link

The link studied is 16 km long, operating at 18.6 GHz, horizontally polarized, oriented NW to SE and fortunately extends almost entirely within the Ze'elim catchment (Fig. 1). The derivation of the rain intensity from the CML data follows the procedure presented, in obedience to the *Power Law*:

$$A_i = a \cdot R_i^b \cdot L \quad \text{(dB)} \tag{2}$$

where $A_i$ is the signal attenuation at time index $i$, $R_i$ (in mm h$^{-1}$) is the instantaneous rain intensity (at time index $i$), $L$ (in km) is the length of the link, and $a$ and $b$ are two parameters, characterized by the frequency, polarization and the rain drop size distribution (Gunn and East, 1954; Leijnse et al., 2007; ITU-R, 1992-1999-2003-2005; Messer and Sendik, 2015). The specific values of parameters $a$ and $b$ are considered to remain relatively constant and were published in the technical literature (ITU-R, 1992-1999-2003-2005). Here, we use data produced by a network management system, obtained by the Israeli cellular provider $Cellcom^{TM}$, which records only the minimum and the maximum TSL and RSL, of 90 samples within a 15 minute interval ($i = 10\ s$). Naturally, the *Power Law* parameters are assigned to an instantaneous attenuation values, but were reevaluated to fit this currently used data storing method ($a = 0.441$, $b = 1.074$), i.e., relating extreme attenuation values to averaged rain intensities (Ostrometzky et al., 2016).

From the minimum TSL (defined by $TSL^{min}$) and the maximum RSL (defined by $RSL^{max}$), the minimum channel attenuation can be approximated:

$$A_j^{min} = TSL_j^{min} - RSL_j^{max} \quad \text{(dB)} \tag{3}$$

where $j$ represents the $j^{th}$ 15 min interval. Similarly, the approximated maximum attenuation can be expressed:

$$A_j^{max} = TSL_j^{max} - RSL_j^{min} \quad \text{(dB)} \tag{4}$$

From the minimum and the maximum attenuation values, the attenuation caused only by rain (defined for the $j^{th}$ interval by $A_j$) can be easily extracted (Ostrometzky and Messer, 2017):

$$A_j = A_j^{max} - min(A_{j-1}^{min}, A_j^{min}) \quad \text{(dB)} \tag{5}$$

which in turn, can be translated directly to a rain-intensity value by using Eq. 2, with the subtraction of $B = 1.6$ dB, shown to be the baseline attenuation, caused by a bias induced by quantization (Ostrometzky et al., 2017):

$$R_j = \sqrt[b]{\frac{A_j - B}{aL}} \quad \text{(mm h}^{-1}) \tag{6}$$

As the length of the link increases, the CML sensitivity to lower rain intensities increases. This characteristic allows the monitoring of small (spatially) and low intensity rain events in rural areas.





## 3.3 Radar

Transmitted microwave pulses are back-scattered to the emitting weather radar as a result of encountering raindrops. By evaluating the power of the back-scattered signal, the radar returns reflectivity ($Z$) values (mm$^6$ m$^{-3}$), more commonly presented in dBZ ($10log_{10}Z$), the convention chosen also for this study. $Z$ is related to rain by the $Z - R$ relations (Krajewski and Smith,

2002), widely examined by Marshall and Palmer (1948); we used the same parameters ($Z = 200R^{1.6}$) as they did. The Israel Meteorological Service (IMS) Doppler corrected C-band radar samples at a 5 min resolution when operated, with a beam width of one degree, elevation of $0.76^o$, and a 125 m radial resolution (Fig. 4). The properties mentioned above converge to create radar cells located roughly $1000\ m$ above ground level with overview dimensions of 125 m by $\approx 1200$ m.

Basic noise reduction measures were undertaken: $Z$ values exceeding 55 dBZ, representing very heavy rain ($\approx 100$ mm

h$^{-1}$), were forced to this value, whereas those not attaining the minimum of 16 dBZ ($\approx 0.36$ mm h$^{-1}$), were regarded as no-rain measurements (Morin and Gabella, 2007). No additional calibration procedures were undertaken, as the aim of this study was to determine the feasibility of using real-time data.

### 3.3.1 ARCOML- Averaged Radar Cells over Microwave Link

Since links in rural areas are generally longer than either dimension of the properties of the radar cell described in the overview,

multiple cells were included to properly represent and for comparison with the line measurement. In an attempt to imitate the link's averaging of rain intensities along the path of the signal, ARCOML, Averaged Radar Cells over Microwave Link, was created: that is, the mean of all radar cells above the link. Here, ARCOML was obtained from five consecutive lateral angles (Fig. 1). In order to match it with the 15 min temporal resolution of the CML, ARCOML was formed by the average of every three (5 min resolution) radar measurements. The temporal compensation was performed only for ARCOML, whereas

in further usage of the radar data, the original temporal resolution was used.

## 4   Rain Spottiness

Differences from the mean value in a series can explain patterns of variation. The standard deviation ($\sigma$), as well as the variance ($\sigma^2$) of the rain intensities along the path of the link, i.e., of all 124 cells constructing the ARCOML, can serve as a measure of rain spottiness along the path.

Another approach, assumed to be more suitable, for determining variability is to examine the distributions' outliers. As the prevalence of higher rain intensities is the point of interest, a moment of higher order was considered. Kurtosis is a measure of how prone to outliers a distribution is, or a measure of the "heaviness of the tail." This fourth order standardized moment (a central moment divided by an expression of $\sigma$) is defined as

$$k = \frac{E(X - \mu)^4}{\sigma^4} \tag{7}$$




where here $n = 124$. In principle, the larger the $k$ is, the heavier the tail of the distribution is (here $k$ of a standard normal distribution equals three). Therefore, the proposed hypothesis is that large $k$ values represent a rain structure that contains more outliers, and as rain intensity values are truncated at zero, outliers are defined as values that are larger than the expected value, under the assumption of positively skewed series. Considering that extreme rain intensities are less prevalent than low ones in a given, sufficiently large spatial (as well as temporal) window, let alone in the Judean Desert, it is reasonable to assume that a snapshot of the intensity-distribution of the rain in a given area (in this case along the 16 km of the CML) will be positively skewed. Moreover, unlike $\sigma$ or $\sigma^2$, kurtosis is affected by neither the multiplication nor the addition of any value to the data, and therefore may be a more robust quantity for the comparison of rain intensities with different amplitudes.

A disadvantage of using $k$ arises as a result of the following property: data series of low and insignificant rain intensity can show kurtosis values identical to those of higher intensity with the same distribution. This property makes it difficult for the naked eye to determine the contribution of $k$ to the spottiness-CMLR relations. Therefore, in order to filter out low rain intensities, a threshold for $k$ calculations was determined: $k$ was not calculated for radar series with ARCOML values not exceeding $0.1 \text{ mm h}^{-1}$.

## 5 Classification Methodology

### 5.1 Effective Rainfall Period

An additional water depth gauging station, Ze'elim Upper, was installed 8.64 km upstream of the gauging station (Fig. 1). Figure 2 shows an example of the calculated discharges. The temporal gap between the two peaks is 50 min, and therefore, based on the aforementioned travel distance, the calculated overall water velocity is $2.88 \text{ m s}^{-1}$. Additionally, sharper slopes occur downstream than upstream of the waterfall (located $9,000$ m upstream of the hydrometric station) (Fig. 3), and therefore, the average velocity along the entire length of the channel is assumed to be somewhat lower. Considering the distance the water travels from the CML-channel crossing point to the gauging station (23.2 km) together with the foregoing, and through a close examination of the progression of the storms, it was determined that the link detects a flash flood generating rain, hereinafter termed an Effective Rain Period (ERP), within a 45 min window, $2.5 - 3.25$ h before the wave is detected at the outlet.

### 5.2 Discharge Rise

Based on the aforementioned, pairs of high spottiness indices and CMLRs of a given ERP were examined with respect to the corresponding hydrological response according to the following procedure:

First, a two sided second order derivative of the discharge hydrograph is conducted

$$\dot{Q}_i = \frac{Q_{i+1} - Q_{i-1}}{2\Delta t} \quad (\text{m}^3 \text{ s}^{-2}) \tag{8}$$

where $\Delta t = 3 \, minute$ is the hydrological sampling resolution for every $i^{th}$ measurement. A single sided derivative would have been too noisy; a two sided one however suppresses the unwanted noise and was therefore chosen. Peaks of $\dot{Q}$ not exceeding



a threshold of 0.003 m$^3$ s$^{-2}$ (representing a change of 1.1 m$^3$ s$^{-1}$) were filtered out. For each peak, the largest associated discharge value was extracted from the window

$$Q_j^{max} = max\left\{Q_{j(i)}, Q_{j(i+1)}, ..., Q_{j(i+3)}\right\} \quad (\text{m}^3 \text{ s}^{-1}) \tag{9}$$

for the $j^{th}$ $\dot{Q}$ peak. Hereafter, the actual hydrological response, $\Delta Q_j^{max}$, is determined by

$$\Delta Q_j^{max} = Q_j^{max} - min\left\{Q_{j(i-7)}, Q_{j(i-6)}, ..., Q_{j(i-1)}\right\} \quad (\text{m}^3 \text{ s}^{-1}) \tag{10}$$

Then, the corresponding ERP is determined, from which pairs of spottiness index-CMLR are coupled as follows. First, the highest CMLR is chosen. Since for every CMLR there are three radar measurements, the highest radar-produced quantity, $k$ in this case, out of the three prior to the CMLR is selected to complete the pairing. If two close $\dot{Q}$ peaks converge to the same hydrograph peak, the pair containing the second highest CMLR in the ERP is chosen. This same methodology was checked for the other two investigated radar products: $\sigma$ and $\sigma^2$.

## 6 Results

Five storms were analyzed that occurred on the following dates: November $6^{th}$, 2015, January $1^{st}$, 2016, January $25^{th}$, 2016, February $22^{nd}$, 2016[1], and January $11^{th}$ 2015.

Prior to the final analysis of the rainfall properties and hydrological responses, a validation of statements regarding the rainfall spottiness is necessary.

### 6.1 Spottiness Validation

Based on correlation between rain gauges (Table 1) that were operating during all five events (Arad IMS and Shani IMS, 16.4 km), the events were sorted from the spottiest to the most uniformly distributed rainfall. Correlations supporting the spottiness classification are shown with two more rain gauges, also located around the CML, e.g., for the spotty November event, the gauge at the greatest distance (Shani IMS) did not monitor rain at all (the gauge was fully operational), whereas the correlation between the gauges closest to one another (Arad and Arad IMS, at a distance of only 3.22 km away) was 0.712. Poor correlation was found at a distance of several kilometers, validating the typical size of rain cells (5 km (Sharon, 1972)). In contrast to the November event, the correlations for the uniform January $25^{th}$, 2016 event were much higher. Table 2 strengths the claim that spottier rain induces higher $k$, where the median, mean and maximal values of $k$ are given in a decreasing order. The spottiness order of the events, based on rain gauge correlations (Table 1), is coherent, except for two events that showed median and mean values that are slightly different from one another.

Figure 6 presents consecutive snapshot histograms of the rain intensities along the CML, as captured by the radar on January $1^{st}$, 2016 during the storm. All the snapshot distributions have positive skewness values. Therefore, it is conceivable that, in

---

[1]Data from a single tower were available for this event, i.e., TSL and RSL were acquired from the same station. Nonetheless, no significant differences are expected, as the TSLs from both towers are similar. The polarization of this channel is vertical, and therefore, the *Power Law* parameters here are $a = 0.419$ and $b = 0.999$





the majority of cases, a rise in $k$ (and probably also in $\sigma$ and $\sigma^2$) is a result of the presence of rain intensities of higher than the expected value. In Table 3, the ARCOML-CMLR correlations are presented. Their fair agreement allows one to relate radar and CML data with higher confidence. It is worth noting that, in the January $11^{th}$, 2015 event, the IMS radar data were intermittently available and therefore the rising part of the hydrograph was unfortunately irrelevant to this study.

## 6.2 Hydrological Responses to Radar-Link Analyzed Data

A storm progression is presented in Fig. 7, where the spottiness index $k$, CMLR, and hydrological response are shown. The points of interest and their corresponding ERPs are marked and respectively numbered. Strong CMLRs are defined within the ERPs, indicating a fair assumption of both the main runoff generating area of the basin and the routing time. ERP's 1 to 3 contain increasing maximal CMLRs, which are associated with $k$s in decreasing order. Further, two consecutive considerably high CMLRs are prominent prior to ERP1 (at 16:00, 16:15 UTC+2). These high rain intensities are suitable for generating a flood, but as transmission losses and the hydraulic conductivity are major factors (especially when the channel is dry), it is conceivable that a flood did form, but infiltrated prior to reaching the gauging station, thereby wetting the channel bed.

Scattered in Figure 8 are all the $k$-CMLR pairs (maximal $k$ for every CMLR) of all five events. Pairs associated with a hydrograph rise are marked by circles of various sizes, representing the coherent $\Delta Q^{max}$. Points that occurred prior to the ERP of the very first water level rise are colored red, whereas those occurring after it are colored blue. A power law model was fitted to the data points related to rises of discharge using least squares. A line representing the regression model minus $RMSD$ is also plotted, indicating an envelope curve, above which the combination of $k$ and CMLR indicate a somewhat higher probability of hydrological responses. The correlation coefficient ($r^2 = 0.627$) and a negative exponential parameter imply an inverse relation. No correlation could be found when the algorithm was tested on $\sigma$ and $\sigma^2$.

## 7 Discussion

Two points were excluded from the regression analysis (marked with a "plus" sign) for the following reasons. It is physically possible that additional data, had they been available, would have followed the regression line further toward lower rain intensities, as a higher expected kurtosis would have indicated a very high intensity somewhere along the line. However, it is possible that at some point, as CMLR increases, the dependency on $k$ weakens as increasingly larger segments of the link exceed the estimated saturated hydraulic conductivity value of the basin, thereby allowing a higher probability that floods will form. A study that dealt with the modeling of a nearby wadi suggested the saturated hydraulic conductivity value is 2 mm h$^{-1}$. Therefore, in Fig. 8 the majority of data points containing CMLR> 2 mm h$^{-1}$ are either associated with a flow or marked as red, meaning the basin could still be rather dry at that point. Therefore, it is reasonable to assume that an upper CMLR threshold (range) exists, after which CMLR-$k$ relations change. It is presumably at this level that $k$ affects only the amplitude of $\Delta Q^{max}$. Moreover, points containing high rain intensities and $k$ values were generated from the spottiest November event, which took place during a different governing synoptic system, the Red Sea Trough, whereas the rest of the events developed



as a result of winter cold fronts. As previously mentioned, Red Sea Trough storms tend to induce highly spotty rain cells, and although considered rare, are held responsible for one third of the flood events in southern Israel.

In spite of the location and orientation of the CML at the uppermost western part of the basin, where the cold fronts approach, some convective cells can develop in an area where the CML has no coverage. Such a case was found in the January $11^{th}$, 2015 event, where a considerable water level rise was noticed, and yet an outstanding reduction in the RSL (not the radar, as the event occurred at the time it was not functioning properly) was noticeable.

## 8   Conclusions

To conclude, the "long isolated Commercial Microwave Link" (CML) scenario examined in this study, used in conjunction with additional information collected by weather radars, is of beneficial value for surface hydrology. It was shown that even when the radar station is located at a great distance, with complex terrain and without calibration procedures, radar data can be used to complement the ground level observations of the CML in determining the ripeness of conditions for flash flood responses. A naive classification approach for the cause of a hydrological response was implemented, where the rain intensity-spottiness relations were tested. The derivative of the discharge hydrograph was used to select sharp discharge rises, as in-basin measurements were used to determine the time window responsible for the runoff generation in the CML vicinity. The relations between the level of spottiness and the Kurtosis ($k$) of radar cells covering the path of the link showed that higher $k$s represent spottier rainfall. No relations were found when statistical moments of lower order were tested. In arid regions worldwide, very few rain gauges exist, but radar observations, although limited, as well as long and sparsely distributed CMLs, can be found even in less populated areas. Therefore, flash flood warning systems can possibly be improved through this approach.

*Code availability.*   TEXT

*Data availability.*   TEXT

*Code and data availability.*   Code is available. Cellular derived rain intensity data is available.

*Competing interests.*   There are no significant competing interests that might have influenced the performance or presentation of the work described in this manuscript.



*Acknowledgements.* We thank Maaian Rotstein for grammar consultation and Dr. Jutta Metzger, Dorian Golriz and Itzhak Lior for technical insights. In $Cellcom^{TM}$ we thank Eli Levi, Yaniv Koriat, Baruch Bar and Idit Alexandrovitz. In the IMS we thank Asaf Rayitsfeld, Elyakum Vadislavsky and Amit Savir. Hydrometric instruments were provided and installed by $Yamma^{TM}$. This work was supported (partly) by the German Research Foundation (DFG) through the project "Integrating Microwave Link Data for Analysis of Precipitation in Complex Terrain:
5  Theoretical Aspects and Hydrometeorological Applications" (IMAP). The German Helmholtz Association is gratefully acknowledged for (partly) funding this project within the Virtual Institute DESERVE (Dead Sea Research Venue) under contract number VH-VI-527.



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





**Table 1.** Correlations ($r$) between all four available rain gauges for the November. $6^{th}$ (**a**), January $1^{st}$ (**b**), January $25^{th}$ (**c**), February $22^{nd}$ (**d**) and January $11^{th}$ (**e**) events. Distances between gauges are presented in parentheses. Unavailable rain gauges are marked as NA. On event **a** the Shani IMS gauge did not monitor rain. Both Arad IMS and Shani IMS gauges (operated by the IMS) have a 10 minute time resolution. Hanokdim and Arad are dedicated gauges which measure every minute. Thus, summation of every 10 samples was used for downscaling.

| **a** Nov. $6^{th}$ 2015 | Arad IMS | Shani IMS | Arad | Hanokdim |
|---|---|---|---|---|
| Arad IMS | 1 | – | 0.712 | 0.065 |
| Shani IMS | (16.40) | – | – | – |
| Arad | (3.22) | (16.93) | 1 | 0.147 |
| Hanokdim | (9.99) | (20.11) | (6.80) | 1 |
| **b** Jan. $1^{st}$ 2016 | | | | |
| Arad IMS | 1 | -0.036 | 0.589 | 0.238 |
| Shani IMS | | 1 | -0.027 | 0.084 |
| Arad | | | 1 | 0.422 |
| Hanokdim | | | | 1 |
| **c** Jan. $25^{th}$ 2016 | | | | |
| Arad IMS | 1 | 0.370 | 0.825 | 0.524 |
| Shani IMS | | 1 | 0.408 | 0.349 |
| Arad | | | 1 | 0.750 |
| Hanokdim | | | | 1 |
| **d** Feb. $22^{nd}$ 2016 | | | | |
| Arad IMS | 1 | 0.628 | 0.812 | NA |
| Shani IMS | | 1 | 0.653 | NA |
| Arad | | | 1 | NA |
| Hanokdim | | | | NA |
| **e** Jan. $11^{th}$ 2015 | | | | |
| Arad IMS | 1 | 0.789 | NA | NA |
| Shani IMS | | 1 | NA | NA |
| Arad | | | NA | NA |
| Hanokdim | | | | NA |





**Table 2.** Classification of the level of storm spottiness using the storm mean and median values of $k$, presented in a decreasing order. Letters in the "Event" column represent the indices given to the storms in the rain gauges spottiness classification in Table 1. The order is coherent except for the exchange of **c** and **d**, which nonetheless, have resembling values.

| Event | Mean | Median |
|-------|------|--------|
| a | 14.2 | 13.5 |
| b | 7.7 | 5.7 |
| **d** | 6.6 | 5.1 |
| **c** | 5.6 | 4.7 |
| e | 4.0 | 3.0 |





**Table 3.** Correlations ($r$) between ARCOML and CMLR for each case study. ARCOML represents the CMLR fairly to very well.

| Event | $r$ |
| --- | --- |
| Nov. $6^{th}$ 2015 | 0.96 |
| Jan. $1^{st}$ 2016 | 0.58 |
| Jan. $25^{th}$ 2016 | 0.83 |
| Feb. $22^{nd}$ 2016 | 0.76 |
| Jan. $11^{th}$ 2015 | 0.89 |





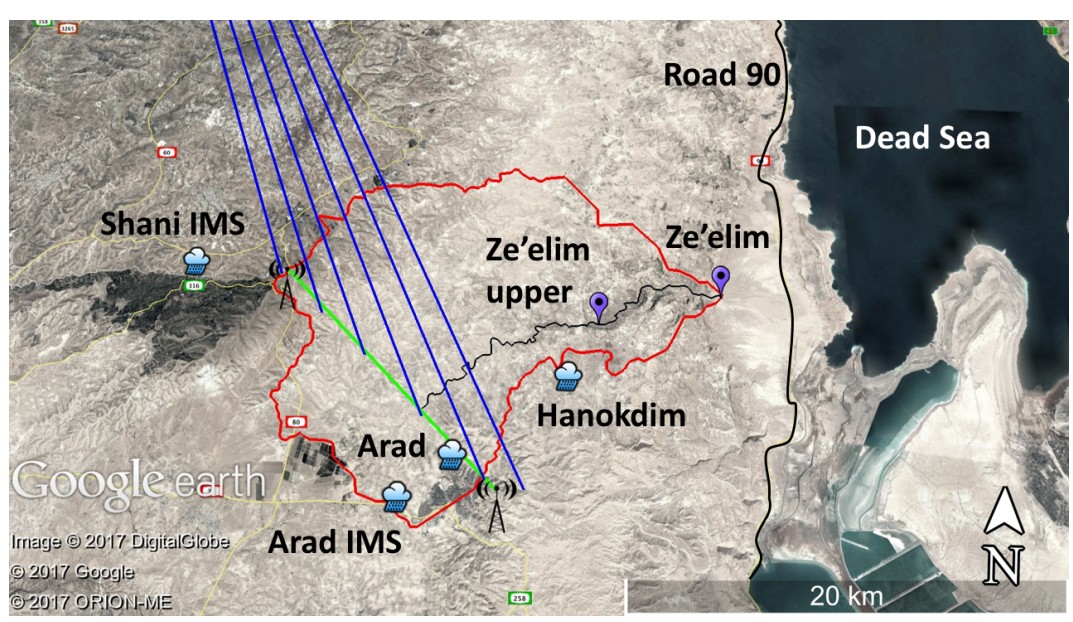

**Figure 1.** Catchment of Wadi Ze'elim (245 km$^2$, red contour), draining into the Dead Sea, crossing road 90. CML across the basin (green), Israeli Meteorological service (IMS) one degree radar beams covering the CML's path (blue), trunk channel (black), rain gauges (light blue markers), Ze'elim and Ze'elim-upper hydrometric stations (purple markers).




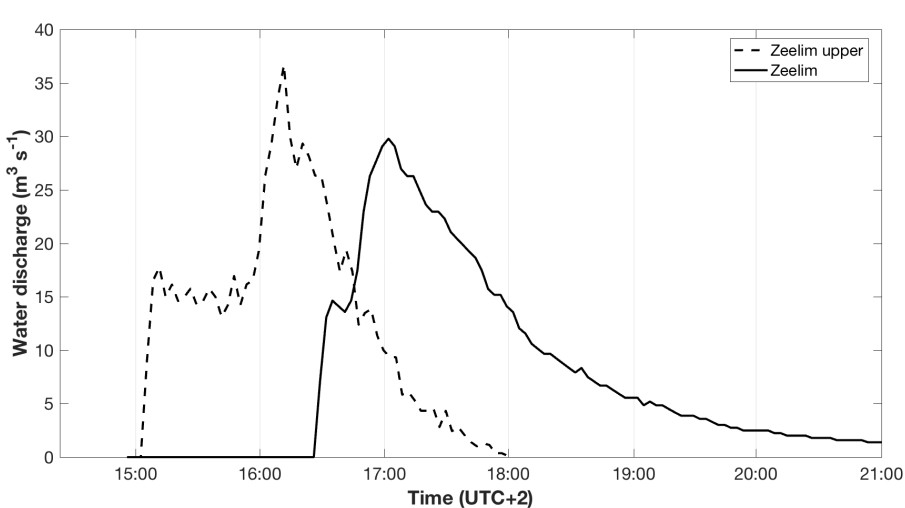

**Figure 2.** Peak-to-peak: 50 min. over 8.64 km between Ze'elim-upper (dashed) and Ze'elim (solid) gauging station (November $6^{th}$ 2015).





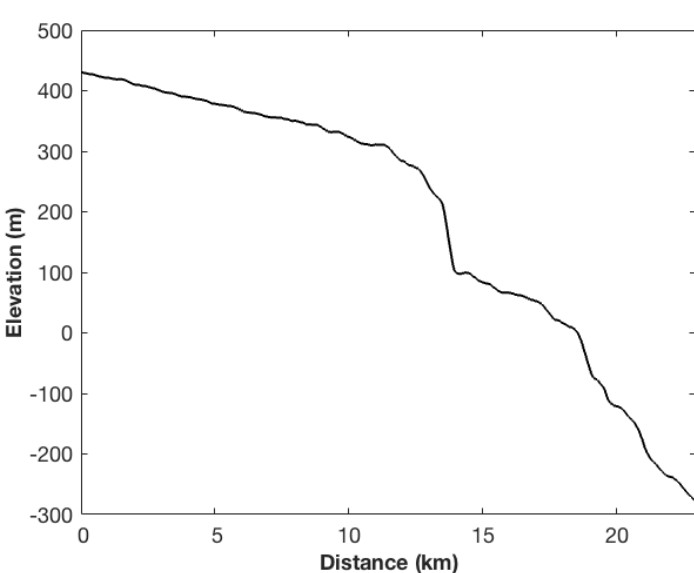

**Figure 3.** Longitudinal profile of the Ze'elim channel. Upper point: CML-channel first crossing point. Lower point: Hydrometric station near road 90.





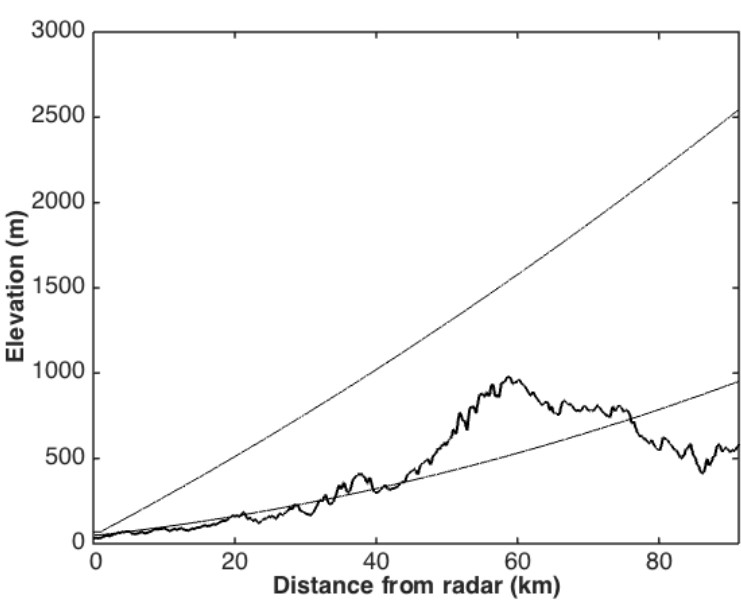

**Figure 4.** Second lowest available radar beam ($0.76^o$). Thin lines represent the top and bottom boundaries of the $1^o$ aperture. Bold line is the topography. The profile is from the IMS by the seashore (Bet-Dagan) to the CML. The Judean Mountains create an obstacle to low radar beams.





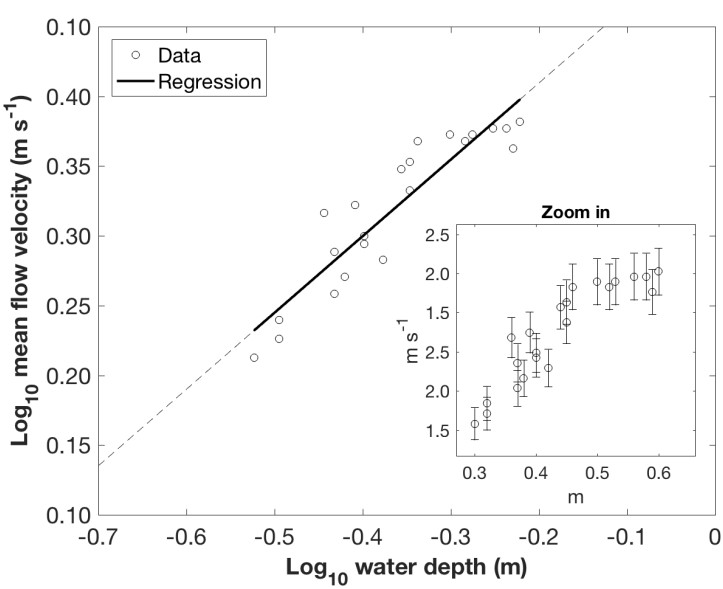

**Figure 5.** Mean surface flow velocity at varying depths. A non logarithmic scale is presented in the small box along with a 5% instrument error. This exceptional data base of water velocity, measured during flash flood events, is used to estimate water discharge. $y = 0.549x + 0.519$; $r^2 = 0.85$.





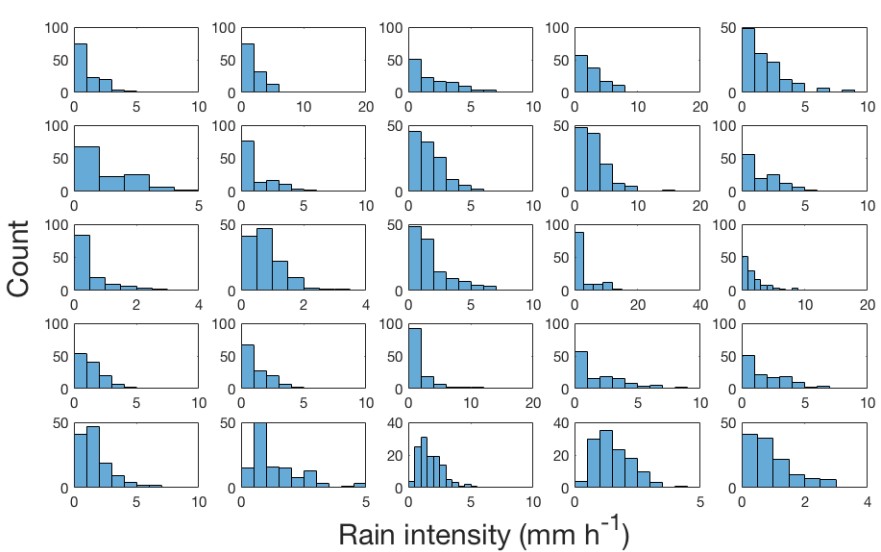

**Figure 6.** Histograms of rain intensities of 124 radar cells on January $1^{st}$ 2016 above the CML, in a 5 minute resolution starting at 05:05 (UTC+2). Low rain intensities are more prevalent and a decaying distribution tail is noticeable.



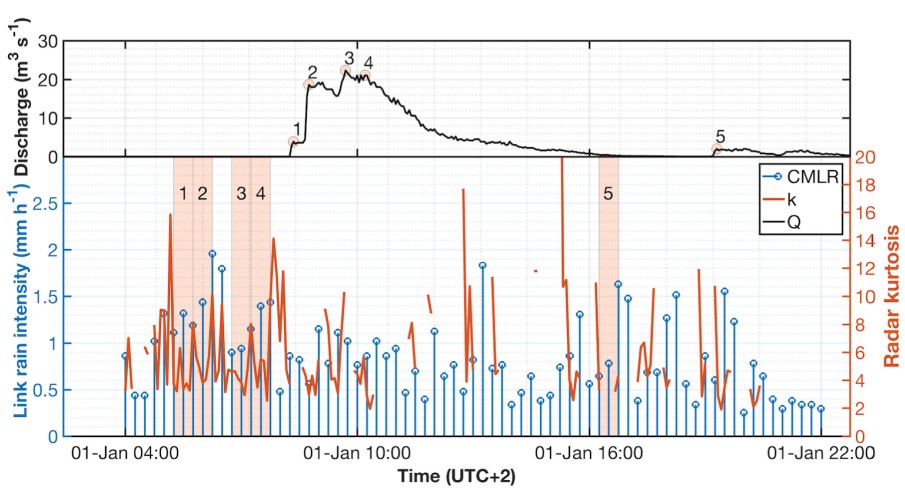

**Figure 7.** Kurtosis spottiness index and rain intensities during Effective Rain Periods (ERPs), with respect to runoff at the outlet of the Wadi. The event presented is January $1^{st}$ 2016. Lower part: CMLR in a 15 minute resolution (blue stems) and radar $k$ in a 5 minute resolution (orange line). Upper part: Discharge hydrograph in Ze'elim outlet (black line). Markers on the hydrograph illustrate detected responses chosen by $\dot{Q}$ and correspond with their ERPs (shaded and numbered areas on the bottom part). Each ERP represents $3.25 - 2.5$ h prior to the marked hydrograph point.





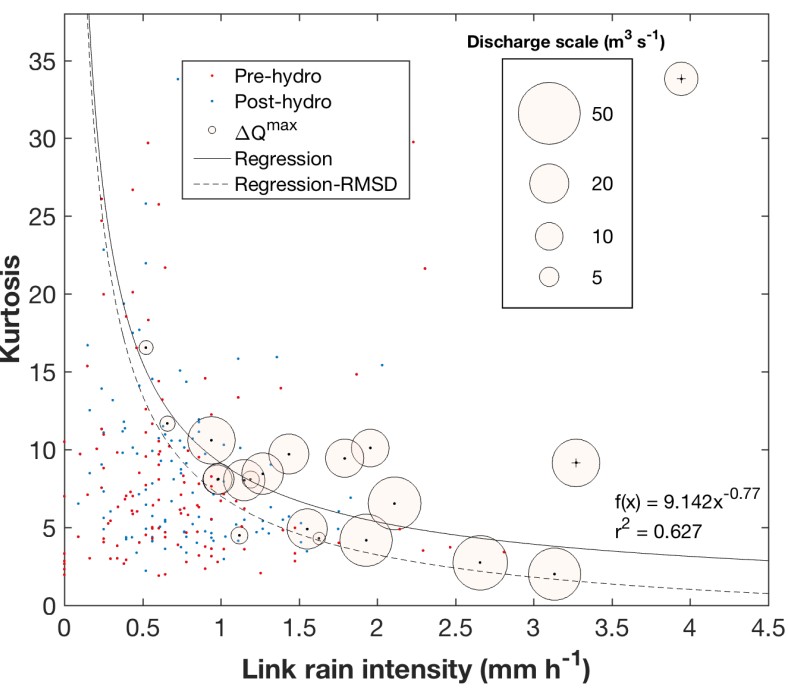

**Figure 8.** $k$-CMLR pairs of all five events. Points chosen by the hydrograph rise algorithm are marked by circles, representing the discharge amplitude. Points which occurred prior (red) and after (blue) the ERP of the first water level are scattered. Power-law regression model (black line) was fitted to the data points chosen by the algorithm. The lower $RMSD$ (dashed line) is presented. Two points were discarded from the regression (plus sign), for reasons detailed in section 7. $RMSD = 2.12$