# Peer review of "On the Use of Measurements from a Commercial Microwave Link for Evaluation of Flash Floods in Arid Regions"

_Atmospheric Chemistry and Physics, 2017_

## Referee Comment (RC1) · Anonymous Referee #1 · 15 Dec 2017

Summary:

A Method for combining microwave link data with radar statistics to identify conditions in which flash floods could develop is presented. The method is applied to a 16-km link and 5 storms over the Wadi Ze'elim catchment in the western part of the Dead Sea rift. Radar is used to quantify the spottiness of the rainfall field while the microwave links are used for rainfall estimation.

Recommendation:

I sincerely doubt that the method proposed by the authors offers added-value for flash flood detection and early warning applications. Most of the paper is speculative in

nature, with little hard evidence to support the conclusions. In particular, I was not convinced by the authors' strong claim that the combination of k (kurtosis) and CMLR leads to useful predictions for flash flood warning and surface hydrology. For example, I do not understand why the commercial microwave link plays such an important role in this story. The same radar data used to derive the kurtosis k along the link could actually be used to derive rainfall estimates as well, probably leading to similar results in terms of flash flood prediction. Sure, maybe the radar-derived rainfall estimates might not be as accurate because of height differences and other radar-related errors. But large radar returns in the region combined with strong spatial variability might still be a pretty solid warning sign for local flooding. So how much do you actually gain by including the microwave link into the method? The paper provides no insight into this, nor does it try to formally assess the usefulness of the method compared with other more traditional ways of doing these things. In summary: the scientific evidence falls short and frankly, does not convince.

Major Comments:

1. Lack of control case: The main goal of the paper is to introduce a new method for combining CML and radar statistics to help identify potentially dangerous conditions for flash floods. The radar is used to assess the spottiness of the rain along a fixed path (using kurtosis k) and CML are used for rainfall intensity estimations. In itself, this is not a bad idea. But how do you demonstrate the value of such an approach? For starter, you need a benchmark against which the proposed technique can be evaluated. Secondly, you need a formal decision rule for distinguishing between dangerous situations and normal ones. None of this is done by the authors. For these reasons, it is impossible to know whether the method has intrinsic value or not. For example, it could be that the CML data are not really improving the detection compared to radar alone. Or conversely, spotiness is not really needed to detect dangerous situations. More evidence and formal testing is needed to support the strong claims made by the authors.

2. No assessment of false positives and false negatives: One important aspect to look at when trying to demonstrate the value of a detection technique is hit rates and false alarm rates. How good is the technique at detecting rises in the hydrograph and how often does it fail? The paper mentions the case of January 11th 2015 where the radar was not working. On this day, a considerable water level rise was noticed but the link did not record any significant attenuation. So maybe the link in itself is not such a good predictor and most of the useful information is coming from the radar? Also, maybe other characteristics derived from radar such as spatial coverage of rainfall over surrounding regions would be more useful than the CMLR?

3. Poorly detailed methodology: the whole methodology for deriving the rain-indued attenuation from minimum and maximum transmitted/received signal levels is sketchy. There are 3 critical parts in the method: (1) the derivation of the baseline and wet antenna attenuation, (2) the removal of the quantization bias due to min/max and (3) the power law transformation. All aspects are poorly explained, with multiple references to non peer-reviewed conference papers. For these reasons, I think it would be good to give more details on the technical aspects of the methods used to retrieve the rainfall from the microwave link.

4. Flawed baseline estimation method: In Section 3.2 Equation (5), the baseline (including wet antenna) proposed here is min(Amin(j-1),Amin(j)), which is a running minimum over the last 2 minimum attenuation values (i.e., corresponding to a time window of 30 min). The authors justify this approach by citing the paper by Ostrometzky and Messer, 2017 (in press). However, this reference turns out to be almost identical to another paper submitted to IEEE Transactions of Geoscience and Remote Sensing back in 2016, which at that time was rejected unanimously by all reviewers (including myself) for its multiple statistical fallacies and methodological weaknesses. It seems like the authors persisted despite the valid criticism and got their flawed paper published almost "as is" in another journal. Back in 2016 when I reviewed this paper, I pointed out that one of the crucial assumptions behind the technique was that the attenuation

measurements needed to be independent from each other. Moreover, the number of samples in the running mean needed to be large enough. Here, the method seems to be applied for the case n=2 (30 min) which, given the temporal dynamics of rainfall, means that two successive attenuation measurements will be highly correlated. As far as I see it, this is a clear violation of the assumptions behind the method. Please justify the approach or choose another more technically sound baseline estimation method.

5. Confusing discussion about outliers and change points: I found Section 7 to be very confusing and speculative. In particular, I could not follow the convoluted arguments given by the authors for justifying why 2 data points were removed from the analysis. Please provide a clearer more solid explanation for this. Moreover, I don't see any strong reason why one could assume that an upper CMLR threshold exists after which the CMLR-k relationship changes. If you think this is the case, please provide hard evidence in the form of an extra statistical analysis of kurtosis-rainfall relationships or give a mathematical derivation supporting the claim. Otherwise, this just looks like you are removing data points that do not support your theory.

6. The quality of the evidence presented in this paper does not support the strong conclusions made by the authors: - For example, the sentence: "The long isolated CML used in conjunction with additional information collected by weather radars is of beneficial value for surface hydrology" is not backed by any data. The evidence you have is circumstantial, showing that pairs of k-CMLR for some selected events are loosely connected to hydrological response. But the relation between the two is not systematic and your analyses do not show you well this works, or how often it fails. This is essential for knowing whether it adds value or not. - Also, the statement that "It was shown that, even when the radar is located at great distance, with complex terrain and without calibration, radar can be used to complement the ground level observations of the CML in determining the ripeness of conditions for flash flood responses." is very misleading. In fact, you do not show any results without the CML. So how can you know whether the combination of CML and radar improves results compared with the

control case where you just consider radar without the CML? Please reformulate or provide a control case where the CML data is not considered. - "Therefore, flash flood warning systems can possibly be improved through this approach." This is speculative. Please remove or show examples of applications where it helped improve flash flood warning.

---

## Short Comment (SC1) · 30 Dec 2017

Dear authors,

as a radar user, I am aware of the limits of the instrument so that seeking for additional information from other sources is largely welcome. In addition, as I got to know the area object of this study, I believe you are addressing an important and difficult challenge in aiming at flash flood warning in the region.

We recently presented the spatial and temporal characteristics of convective rainfall in the same area (https://doi.org/10.1016/j.atmosres.2017.09.020) using a high resolution

[Figure]

X-Band radar benchmarked with data from the C-Band radar you use, even though with an improved elaboration procedure. We analyzed 11 events, including at least one of the events also presented in this study, finding very large spatial and temporal variability: correlation distances < 5 km, time-correlation distances < 10 min.

This raises questions about this study: how can 30-min measurements of average attenuation along a 16-km link covering part of the catchment provide information for flash flood warning in the Ze'elim basin without the radar information? Such a microwave link cannot provide information on (a) the local rain intensities occurring within the 16-km link – this length is much larger than the typical scales of convective rainfall in the area, and this is partially addressed by the method you propose, and (b) all for the portions of the catchment not covered by the link path.

The method you propose makes use of radar data to add indirect information on spatial rainfall variability to the link's quantitative estimates. Information from both instruments is thus required, but this information is only partially exploited. In particular, radar data is believed to represent the spatial variability of rain, but it is not trusted quantitatively since rain gauges cannot be reliably used to adjust the radar data in these conditions of spatial variability. However, the microwave link provides local quantitative information aggregated over a 16-km path, thus more reliable than that of rain gauges.

Why not using the link to adjust the radar estimates? This would provide spatially distributed information over the full catchment, improving the quantitative accuracy of radar estimates and fully exploiting the characteristics of the two instruments. At this point, both kurtosis and quantitative rainfall estimates from the radar over the full catchment could be used following the method you are proposing.

Best regards,
Francesco Marra

---

## Referee Comment (RC2) · Anonymous Referee #2 · 6 Jan 2018

Review: **On the Use of Measurements from a Commercial Microwave Link for Evaluation of Flash Floods in Arid Regions**

The paper by Eshel et al proposes an integration of commercial microwave links and weather radar data to improve understanding of flash flood generation and potentially use for flash flood warning. The authors suggest an innovative approach to consider kurtosis of radar rainfall along a link together with the CML-estimated rain rate (representing the mean along the link). The former represents the spottiness of rainfall that is an important property for desert flash flood generation, together with the mean rain rate.

In general, a "smart" integration of rainfall data from different sensors, to better understand and predict flash floods should be scientifically encouraged. But, the underline assumption in these approaches is to build on the larger strength of each sensor. Knowing the large sensitivity of flash floods (and in particular desert flash floods) to rainfall spatial variability it is hard to understand why the proposed integration approach does not utilize the radar rainfall spatial distribution over the catchment and use the more accurate CML rainfall estimate to correct the mean rainfall bias? The main advantage of the radar rainfall is their spatial distribution and full coverage of the catchment while the main advantage of the CML rainfall is its accuracy and its mean areal nature (as opposed to point data from gauges). So, correcting the radar bias with the CML data seem as the most reasonable way to go. The authors should explain why did they choose the specific approach presented.

In addition, the scientific message of the main result of the paper, i.e., the k-CMLR relations (Figure 8), is not clear. Is the main point here the negative high correlation exists between mean areal rainfall and kurtosis for flood producing storms? or is it the envelope curve suggesting that for a given mean rain intensity (CMLR) one can identify a threshold kurtosis that supports flash flood generation? These are two different things. If the first one is the main message – this is a nice result (but must be more carefully checked and especially understand the kurtosis nature), but it is not related to flood prediction. If the second – then the high correlation is not an issue. Also, looking at figure 8 it seems that most of the circles are right to 1 mm/h (there are some points without circles with larger rain intensity but also there are quite few such points above the envelope curve), so – does the information about the kurtosis really improve prediction?

Specific comments:

Sources of errors in CML rainfall estimates: it would be beneficial to give the reader the sources of errors in the introduction section (1.1). Also, if possible please provide some quantitative information about the typical errors. For example, "CMLs can provide a fair ground truth for rain in populated areas, where the networks are denser." – it would be good to give the values of errors from this analysis.

P. 3 line 21-26: The authors write that "The approach is a complementary integration, using the advantages of each rain monitoring instrument to compensate the weaknesses of the other, with respect to the hydrological responses measured…". I tend not to agree. The weakness of the CML is that it does not cover the entire catchment and also provides too coarse resolution data; its

advantage is the higher accuracy. The opposite for the radar data. How does the suggested approach use the advantages of each method to compensate the weaknesses of the other? To me it is not clear.

P. 4 L 10: "This implies that most of the annual rainfall is recorded by gauges located in the western part of the basin, as is the studied CML.". But not necessarily most of the flood producing rainfall is upstream, because of the lower infiltration rates at the downstream part.

P. 4 L 27: Discharge estimation: it is not clear how from two velocity measurements one can derive the discharge of the full hydrograph for the five events. Please clarify.

Figure 5: seems not to be referred in the text.

P. 5 L 13: how are the parameters given (a and b) different from the published parameter values for this configuration (wave length, etc.)? are you sure the only cause of these different parameter values is the use of min and max data rather than continuous data?

Kurtosis: rain rates have typically very skewed distributions. How well does the kurtosis parameter describe the heaviness of the tail for skewed distributions (as opposed to normal distributions)? Is it independent of skewness? What other parameters were proposed in the literature to describe tail thickness?

Section 5: this section is not clear - why is classification needed? Classification of what? Please clarify what is the goal of the methodology described in this section and its rational.

P. 7 L. 18: the velocity given is the wave celerity rather than the water velocity.

P. 7 L. 9: this is an approximation of discharge derivative.

P. 7 L. 30: why to consider discharge derivative? Please provide the rational.

P. 8 L. 18: rain gauges are used to check the storm spottiness, but this can suffer from all the problems related to point representation of the storm that are well known. Why not use the radar data instead?

Figure 8 presents red and blue points indicating different wetness conditions. I would expect the author to check if these two data sets present any (statistically significant) different behavior. Such a difference is not clear from the visual inspection of the figure.

What is the message in Figure 8 (see my major comment above)? The authors must better clarify it.

Discussion section: reading this section I feel it should be the continuation of the previous section showing figure 8. The text in section 7 refers mainly to the results shown in this figure. This is not a standard discussion section where more general issues are raised and the results of the present study are discussed with relation to other studies. I suggest to combine this part into Section 6.

Minor comments:

P. 2 L8 – state the typical spatial resolution of radar

P. 2 L11 – "Despite"

P. 3 L. 10: "Rainfall events, which occur several times a year, frequently generate flash floods in the region" – it is unclear what this sentence actually states.

P. 3 L. 30: Soils should also be described.

P. 4 L. 2: No need of a minus sign if you write "below sea level".

P. L. 26: I Suggest to change the sign for wetted flow area. The "f" looks like a power.

P. 5 L. 27: The word "rural" seems not appropriate here.

P. 6 L. 6: Why using Marshal-Palmer relations? They are more appropriate to stratiform rain.

P. 6 L. 8: The height given is for the study area.

P. 7 L. 16: The second station should be mentioned earlier in the study area description.

P. 9 L. 26: Please provide a reference to the saturated hydraulic conductivity of 2 mm/h.

---

## Author Comment (AC1) · 28 Jan 2018

First, we would like to thank anonymous referee #1 for his review.

Comment #1: Lack of control case: The main goal of the paper is to introduce a new method for combining CML and radar statistics to help identify potentially dangerous conditions for flash floods. The radar is used to assess the spottiness of the rain along a fixed path (using kurtosis k) and CML are used for rainfall intensity estimations. In itself, this is not a bad idea. But how do you demonstrate the value of such an approach? For starter, you need a benchmark against which the proposed technique can be evaluated. Secondly, you need a formal decision rule for distinguishing between

dangerous situations and normal ones. None of this is done by the authors. For these reasons, it is impossible to know whether the method has intrinsic value or not. For example, it could be that the CML data are not really improving the detection compared to radar alone. Or conversely, spotiness is not really needed to detect dangerous situations. More evidence and formal testing is needed to support the strong claims made by the authors.

Response to comment #1: We thank anonymous referee #1 for the careful compliment that our approach "is not a bad idea". Indeed, our new method is not an attempt to introduce the method as an on-the-shelf ready method, most importantly because the link data are presently unavailable online, whereas the ability to identify dangerous situations requires online availability, such as rainfall recorder and radar backscatter data. Rather, we introduce a concept that requires further development for implementation. Having said that, both the concept and the methodology are novel. As such, we contend that they are eligible to be brought to the attention of the public.

It has previously been shown that estimating rainfall by the use of CML is accurate, but the "long isolated link challenge" has not yet been considered. Doing so in an area where rain cells are much smaller than the link's path emphasizes the challenge. The uniqueness of the studied area largely relies on highly spotty rainfall and strong, very localized rain bursts, characteristics which were aimed to be highlighted as they play a major role in desert surface hydrology. The study initially involved no radar data, but as integrated rain intensities along 16 km failed in providing sufficient information with regard to runoff response, we included the distribution along the link.

The challenge dealt with in this study is the insufficiency of radar in remote, dry mountainous areas as well as that of long CML when exposed to the aforementioned rain characteristics. Thus, consideration of the spatial distribution (rather than the quantities) of radar cells in combination with ground level observations (which have their own limitations) brings forth the strengths of each instrument. This is the justification.

Comment #2: No assessment of false positives and false negatives: One important aspect to look at when trying to demonstrate the value of a detection technique is hit rates and false alarm rates. How good is the technique at detecting rises in the hydrograph and how often does it fail? The paper mentions the case of January 11th 2015 where the radar was not working. On this day, a considerable water level rise was noticed but the link did not record any significant attenuation. So maybe the link in itself is not such a good predictor and most of the useful information is coming from the radar? Also, maybe other characteristics derived from radar such as spatial coverage of rainfall over surrounding regions would be more useful than the CMLR?

Response to comment #2: Miss detect and false alarm cases, by which one can effectively determine the contribution of the approach, will add to the solidity of our approach. The reasons for the absence of such tests are as follows: Data scarcity. As significant rain events in the region are rare (very few a year, if at all) there are simply insufficient data at this stage. For instance, Wadi Ze'elim has been monitored merely since 2015. In addition, finding a hydrologically monitored catchment with links in and/around its basin which store data in a suitable format (in our case fully dependet on the cellular services provider) is impossible in most locals. It has been shown that real-time, countrywide data possession is possible (Chwala et al., 2015). Therefore, at this stage the manuscript aims to demonstrate an idea which can potentially be developed for application.

Weather radars do provide spatial coverage, as limited as they are in these areas; these were obviously used only over the CML and not over the entire catchment. The contributions and limitations of radar in catchment hydrology are well known. Nevertheless, to demonstrate the triangular connection: radar-CML-flash flood, as a "concept demonstration", our agenda is to put the CML itself and the variations within it's path in focus, and thereby to extrapolate to the general case in the future. The suggestions proposed by the referee should be included in a future study which, albeit the difficulties should include additional catchments, more CMLs and additional focus on utilization of

radar capabilities.

Comment #3: Poorly detailed methodology: the whole methodology for deriving the rain-indued attenuation from minimum and maximum transmitted/received signal levels is sketchy. There are 3 critical parts in the method: (1) the derivation of the baseline and wet antenna attenuation, (2) the removal of the quantization bias due to min/max and (3) the power law transformation. All aspects are poorly explained, with multiple references to non peer-reviewed conference papers. For these reasons, I think it would be good to give more details on the technical aspects of the methods used to retrieve the rainfall from the microwave link.

Response to comment #3: We thank the referee for illuminating the need for further detailed explanations. This chapter will be expanded. Concerning the quality of the relevant cited conference papers, all are full-length, peer-reviewed conference proceedings. In particular, ICASSP (International Conference on Acoustics, Speech, and Signal Processing) is a major IEEE sponsored conference, and is the main annual venue for the IEEE SPS (Signal Processing Society). The ICASSP proceedings rank as the 4th top publication for the signal-processing category, with an H5 index of 67 (https://scholar.google.co.il/citations?view_op=top_venues&hl=en&authuser=1&vq=eng_signalprocessing).

Comment #4: Flawed baseline estimation method: In Section 3.2 Equation (5), the baseline (including wet antenna) proposed here is $\min(A_{min}(j-1), A_{min}(j))$, which is a running minimum over the last 2 minimum attenuation values (i.e., corresponding to a time window of 30 min). The authors justify this approach by citing the paper by Ostrometzky and Messer, 2017 (in press). However, this reference turns out to be almost identical to another paper submitted to IEEE Transactions of Geoscience and Remote Sensing back in 2016, which at that time was rejected unanimously by all reviewers (including myself) for its multiple statistical fallacies and methodological weaknesses. It seems like the authors persisted despite the valid criticism and got their flawed paper published almost "as is" in another journal. Back in 2016 when I reviewed this paper,

I pointed out that one of the crucial assumptions behind the technique was that the attenuation C3 ACPD Interactive comment Printer-friendly version Discussion paper measurements needed to be independent from each other. Moreover, the number of samples in the running mean needed to be large enough. Here, the method seems to be applied for the case n=2 (30 min) which, given the temporal dynamics of rainfall, means that two successive attenuation measurements will be highly correlated. As far as I see it, this is a clear violation of the assumptions behind the method. Please justify the approach or choose another more technically sound baseline estimation method.

Response to comment #4: This is a very interesting topic of using CMLs in different climate regions. n=2 (last 30 minutes) as a dynamic window of the minimum attenuation might be more debatable in high latitudes regions, but in a rain shadow area it is reasonable, to our opinion, especially with the very high spatial as well as temporal variability. Moreover, it is mentioned that the min max levels are constructed based on 10 s measurements. Thus, theoretically the max values of two successive measurements can be very close time wise but also 30 min apart, which makes them highly non correlative with respect to desert meteorology. We also tried n=8, with no significant difference detected from n=2. Regardless, additional methods of derivation of rain intensities were attempted, including: (a) subtraction of the median of the average values of min&max received signal levels in the past 24 h (Overeem et al., 2011), and (b) subtracting a 2.3 dB wet antenna according to the procedure detailed in Overeem et al., 2013 (including a local optimization of alpha). The chosen method appears to be more suitable for the area, most likely due to the different climatologic character than in the aforementioned studies. But finally, the rain derivation method is not the scope of the paper. The rain intensity in this study is a tool of demonstrating the focal point, so as long as it is derived by published methods, it serves the purpose. Discourse on rain derivation methods is important. Therefore, a quantitative comparison should definitely be conducted in the future.

Comment #5: Confusing discussion about outliers and change points: I found Section

7 to be very confusing and speculative. In particular, I could not follow the convoluted arguments given by the authors for justifying why 2 data points were removed from the analysis. Please provide a clearer more solid explanation for this. Moreover, I don't see any strong reason why one could assume that an upper CMLR threshold exists after which the CMLR-k relationship changes. If you think this is the case, please provide hard evidence in the form of an extra statistical analysis of kurtosis-rainfall relationships or give a mathematical derivation supporting the claim. Otherwise, this just looks like you are removing data points that do not support your theory.

Response to comment #5: Indeed, an interesting comment. The two "outlier points" are a product of the same event, in fact the spotty most event. We are thankful to possess these points as they strengthen the suggested theory that a further research (when sufficient data are available) should approach this issue using envelope curves or pattern recognition / non linear logistic regression, to state thresholds.

The claim of a threshold beyond which the CMLR-k relations loosen, in regard to runoff generation, derives from the physics of infiltration rates and ponding times. For the general case, a rain intensity lower than the lowermost value of the hydraulic conductivity of a given soil, will not generate runoff. When rain intensity is averaged over 16 km, this "lower value" doesn't imply that there are no areas along the line which exceed this value. This is where kurtosis comes in. For average rain rate values exceeding the hydraulic conductivity for minimal runoff generation, the distribution along the link plays a minor role (at least when classifying floods binarily) as it is theoretically possible that somewhere along the path the rain intensity is sufficiently high to generate runoff, regardless of the spatial distribution.

Comment #6: The quality of the evidence presented in this paper does not support the strong conclusions made by the authors: - For example, the sentence: "The long isolated CML used in conjunction with additional information collected by weather radars is of beneficial value for surface hydrology" is not backed by any data. The evidence you have is circumstantial, showing that pairs of k-CMLR for some selected events are

loosely connected to hydrological response. But the relation between the two is not systematic and your analyses do not show you well this works, or how often it fails. This is essential for knowing whether it adds value or not. - Also, the statement that "It was shown that, even when the radar is located at great distance, with complex terrain and without calibration, radar can be used to complement the ground level observations of the CML in determining the ripeness of conditions for flash flood responses." is very misleading. In fact, you do not show any results without the CML. So how can you know whether the combination of CML and radar improves results compared with the control case where you just consider radar without the CML? Please reformulate or provide a control case where the CML data is not considered. - "Therefore, flash flood warning systems can possibly be improved through this approach." This is speculative. Please remove or show examples of applications where it helped improve flash flood warning.

Response to comment #6: We acknowledge that the sentence: "The long isolated CML used in conjunction with additional information collected by weather radars is of beneficial value for surface hydrology" will be written in a less confident manner.

Regarding the sentence: "It was shown that, even when the radar is located at great distance, with complex terrain and without calibration, radar can be used to complement the ground level observations of the CML in determining the ripeness of conditions for flash flood responses". We disagree with the referee's statement that this sentence is "very misleading". The horizontal axes of Fig. 8 demonstrate that the CMLR alone provides very little information about the expected hydrologic response, mostly due to it's length. It is prominent that the addition of radar data enhances the CML observations. Furthermore, and as shown, there is at least a 2 hour gap between the CML detection until the response of the river occurs at the outlet. This time window is crucial for people living in such areas, when preparedness at short notice is crucial. Calibrating radar data is time consuming, and using quantitative values obtained by it in such an area is insufficient, finely summarized by Brene and Krajewski, 2013.

[Figure]

Regarding the sentence: "Therefore, flash flood warning systems can possibly be improved through this approach." The word "possibly" makes this sentence speculative in nature. Nonetheless, this is our speculation, and we will amend or remove the sentence if necessary.

References

Chwala et al., 2015.

Overeem et al., 2011.

Overeem et al., 2013.

Brene and Krajewski, 2013

---

## Short Comment (SC2) · 6 Feb 2018

In thanking the authors for the response, I want to stress that my comment is not a review report and should not be considered as a complete review of the manuscript neither by the authors nor by the editor. It is a "short comment", within the open discussion context offered by this journal, stimulated by the discussion manuscript.

Best regards, Francesco Marra

---

## Author Comment (AC2) · 6 Feb 2018

First, we would like to thank Dr. Marra for his review.

Referee comment: This raises questions about this study: how can 30-min measurements of average attenuation along a 16-km link covering part of the catchment provide information for flash flood warning in the Ze'elim basin without the radar information? Such a microwave link cannot provide information on (a) the local rain intensities occurring within the 16-km link – this length is much larger than the typical scales of convective rainfall in the area, and this is partially addressed by the method you propose, and (b) all for the portions of the catchment not covered by the link path.

[Figure]

Response: Thank you for the opportunity given to clarify our aim. We agree that a single, long link cannot be effectively used for flash flood warning with the absence of side information, especially due to the two reasons you mentioned. In regard to the first (a), the aim to reveal which kind of rain patterns induce types of attenuation was indeed the reason for the additional use of radar data. You are correct with regard to comment (b). We are aware that convective cells can develop in the eastern parts of the basin (even if this is rare) and as there are no links in that area we are limited. Data fusion can go further with additional use of the spatial capabilities of radar, preliminary calibrated products and more. Nonetheless, we chose to focus on a very specific aspect: the effect of the spatial distribution along the link on the hydrologic response to demonstrate the proposed method. We possess data of other links in the area (but not directly in the basin), several rain gauges (today), and the X band radar you mention, which can thereby be used for future research.

Referee comment: Why not using the link to adjust the radar estimates? This would provide spatially distributed information over the full catchment, improving the quantitative accuracy of radar estimates and fully exploiting the characteristics of the two instruments. At this point, both kurtosis and quantitative rainfall estimates from the radar over the full catchment could be used following the method you are proposing.

Response: Indeed, an interesting topic. This subject has already been studied (e.g., Cummings et al., 2009), but nonetheless, further progress can be (and perhaps should) be undertaken with regard to the implementation into catchment hydrology.

References Cummings et al., 2009

---

## Author Comment (AC3) · 12 Feb 2018

We would like to thank anonymous referee #2 for his review.

Comment: In general, a "smart" integration of rainfall data from different sensors, to better understand and predict flash floods should be scientifically encouraged. But, the underline assumption in these approaches is to build on the larger strength of each sensor. Knowing the large sensitivity of flash floods (and in particular desert flash floods) to rainfall spatial variability it is hard to understand why the proposed integration approach does not utilize the radar rainfall spatial distribution over the catchment and use the more accurate CML rainfall estimate to correct the mean rainfall bias? The

main advantage of the radar rainfall is their spatial distribution and full coverage of the catchment while the main advantage of the CML rainfall is its accuracy and its mean areal nature (as opposed to point data from gauges). So, correcting the radar bias with the CML data seem as the most reasonable way to go. The authors should explain why did they choose the specific approach presented.

Response: A similar topic was raised by Francesco Marra earlier in the current discussion, hence our response may contain some overlaps. This subject was studied in the past (e.g., Cummings et al., 2009) in a relatively well gauged area. Further progress can be done in adjusting the rainfall field as well as the implementation to flood applications. In this study we chose to focus on a very specific aspect: the effect of the spatial distribution along the link on the hydrologic response. This aspect, apart from providing a better understanding of the CML "behavior" when encountering different rain patterns, can constitute an important first step of a future study which will include your suggestions in a non- well gauged, semi-arid area. We believe the fundamental understanding of the advantages, limitations and the CML rain rate-runoff response are crucial as a first step.

Comment: In addition, the scientific message of the main result of the paper, i.e., the k-CMLR relations (Figure 8), is not clear. Is the main point here the negative high correlation exists between mean areal rainfall and kurtosis for flood producing storms? or is it the envelope curve suggesting that for a given mean rain intensity (CMLR) one can identify a threshold kurtosis that supports flash flood generation? These are two different things. If the first one is the main message – this is a nice result (but must be more carefully checked and especially understand the kurtosis nature), but it is not related to flood prediction. If the second – then the high correlation is not an issue. Also, looking at figure 8 it seems that most of the circles are right to 1 mm/h (there are some points without circles with larger rain intensity but also there are quite few such points above the envelope curve), so – does the information about the kurtosis really improve prediction?

Response: Thank you for bringing up this issue. We also grappled with the envelope curve. As the data are so rare we decided that claiming such a curve would be too early, but nonetheless, a dedicated curve methodology perhaps should be reconsidered. First, we intend to claim the connection is sealed by a lower curve. As we used quite a sensitive threshold for the hydrologic response ($\Delta$=1 m3 s-1), we consider the captured data points represent a lower bound to the "real" envelope, which we claim exists.

Specific comments:

Comment: Sources of errors in CML rainfall estimates: it would be beneficial to give the reader the sources of errors in the introduction section (1.1). Also, if possible please provide some quantitative information about the typical errors. For example, "CMLs can provide a fair ground truth for rain in populated areas, where the networks are denser." – it would be good to give the values of errors from this analysis.

Response: Sources of error can certainly be added.

Comment: P. 3 line 21-26: The authors write that "The approach is a complementary integration, using the advantages of each rain monitoring instrument to compensate the weaknesses of the other, with respect to the hydrological responses measured. . .". I tend not to agree. The weakness of the CML is that it does not cover the entire catchment and also provides too coarse resolution data; its advantage is the higher accuracy. The opposite for the radar data. How does the suggested approach use the advantages of each method to compensate the weaknesses of the other? To me it is not clear.

Response: The approach aims to enhance the contribution of the CML, is in its inner rain rate variability. You tend not to agree, but in our perception of your comment it is either that you do agree or that the complementary point of view was not well understood, and therefore needs to be amended. The claim is that the spatial resolution of the link is not sufficiently good, but at the same time constitutes a ground truth, whereas the

radar is not accurate quantitatively but reveals the spatial distribution. We utilize both aforementioned strengths.

Comment: P. 4 L 10: "This implies that most of the annual rainfall is recorded by gauges located in the western part of the basin, as is the studied CML.". But not necessarily most of the flood producing rainfall is upstream, because of the lower infiltration rates at the downstream part.

Response: This was mentioned in the description of the basin in the "Study region" chapter, but will be amended to be more specific. The nature of the rain shadow is that the majority of the storms approach from the west, and thus, all the events chosen for this analysis are such. Of course, different approaches and more CMLs from the vicinity can be used to cover a larger area, thereby leaving less blind spots. But, this additional aspect belongs to the next step of this research, which should be a Proof of Concept, including more links, spatial observations, and methods of calibration to determine how the radar can be made more trustworthy where CMLs are not present. At the moment we are in the stage of demonstrating the concept, therefore such details can not be covered as yet.

Comment: P. 4 L 27: Discharge estimation: it is not clear how from two velocity measurements one can derive the discharge of the full hydrograph for the five events. Please clarify.

Response: Water velocity measurements were possible during two flood events (a considerable achievement considering the remote location, the rare occurrence of flash floods and their limited predictability). In both cases, velocities at various depths were measured, thereby forming Figure 5, from which the the depth-velocity curve was derived.

Comment: Figure 5: seems not to be referred in the text.

Response: Thank you. Will be added in P. 4 L 28.

Comment: P. 5 L 13: how are the parameters given (a and b) different from the published parameter values for this configuration (wave length, etc.)? are you sure the only cause of these different parameter values is the use of min and max data rather than continuous data?

Response: The power law parameters are published by the ITU (ITU P.838-3). From there, the specific a and b parameters for a given configuration (i.e., the frequency and the polarization of the signal) can be taken. However, these parameters are based on the instantaneous power law (meaning - that an instantaneous rain intensity value (in mm/h) at a given time (t) will cause a path-attenuation at the same time (t) (in dB/km), which can be calculated using a and b). In our case, we merely have access to the minimum and the maximum TSL and RSL values at 15-minute intervals, from which the approximated minimum and maximum attenuation values within the 15-minute interval can be calculated. It was recently shown that the same power law-like relationship can be used to relate the averaged rain-intensity within the 15-minute interval with the maximum (or minimum) attenuation value, by taking a calibrated "a" parameter (Ostrometzky et al., 2016). The calibrated "a" parameter is indeed a function of the original published ITU "a" parameter, and thus, it remains a function of the specific signal frequency and polarization.

Comment: Kurtosis: rain rates have typically very skewed distributions. How well does the kurtosis parameter describe the heaviness of the tail for skewed distributions (as opposed to normal distributions)? Is it independent of skewness? What other parameters were proposed in the literature to describe tail thickness?

Response: Concerning symmetric distributions, the odd cumulants are zero whereas the even ones, for k>2, represent a change from the gauss distribution. The kurtosis is a fourth order cumulant which is highly important as it contains the information about the manner of the change. There can be additional data about the tails in cumulants of higher order but their calculation is rather cumbersome and inserts errors, and moreover, their contribution is generally negligible.

Comment: Section 5: this section is not clear - why is classification needed? Classification of what Please clarify what is the goal of the methodology described in this section and its rational.

Response: The classification relates to the description of which hydrological responses and their k-CMLR pairs are chosen eventually. The aforementioned is the goal of this technical section.

Comment: P. 7 L. 18: the velocity given is the wave celerity rather than the water velocity.

Response: Thank you for the correction.

Comment: P. 7 L. 30: why to consider discharge derivative? Please provide the rational.

Response: As hydrographs, especially in this area, are reacting with sharp increasing discharges, the discharge derivative points directly on a response to high rain intensity.

Comment: P. 8 L. 18: rain gauges are used to check the storm spottiness, but this can suffer from all the problems related to point representation of the storm that are well known. Why not use the radar data instead?

Response: Rain gauges du suffer from the above but are nonetheless acceptable for this purpose. Moreover, as we are using the radar to represent the spottiness nevertheless (in the form of kurtosis), we wanted to provide a proof to our claim by another instrument.

Comment: Figure 8 presents red and blue points indicating different wetness conditions. I would expect the author to check if these two data sets present any (statistically significant) different behavior. Such a difference is not clear from the visual inspection of the figure. What is the message in Figure 8 (see my major comment above)? The authors must better clarify it.

Response: The division to blue and red points was mainly to strengthen the claim that the envelope curve is valid only only up to a certain rain intensity value which should be close to the saturated hydraulic conductivity coefficient of the basin. It is described in detail in our opinion but can be further explained if necessary.

Comment: Discussion section: reading this section I feel it should be the continuation of the previous section showing figure 8. The text in section 7 refers mainly to the results shown in this figure. This is not a standard discussion section where more general issues are raised and the results of the present study are discussed with relation to other studies. I suggest to combine this part into Section 6.

Response: Thank you. This can be done if necessary.

References: Cummings et al., 2009 Ostrometzky et al., 2016